# Surveillance for Severe Acute Respiratory Infections among Hospitalized Subjects from 2015/2016 to 2019/2020 Seasons in Tuscany, Italy

**DOI:** 10.3390/ijerph18083875

**Published:** 2021-04-07

**Authors:** Ilaria Manini, Andrea Camarri, Serena Marchi, Claudia Maria Trombetta, Ilaria Vicenti, Filippo Dragoni, Giacomo Lazzeri, Giovanni Bova, Emanuele Montomoli, Pier Leopoldo Capecchi

**Affiliations:** 1Department of Molecular and Developmental Medicine, University of Siena, Via Aldo Moro 2, 53100 Siena, Italy; serena.marchi2@unisi.it (S.M.); trombetta@unisi.it (C.M.T.); giacomo.lazzeri@unisi.it (G.L.); emanuele.montomoli@unisi.it (E.M.); 2Interuniversity Research Center on Influenza and Other Transmissible Infections (CIRI-IT), Via Pastore 1, 16132 Genoa, Italy; 3Emergency and Transplants Department, University Hospital of Siena, Viale Bracci 16, 53100 Siena, Italy; a.camarri@ao-siena.toscana.it (A.C.); g.bova@ao-siena.toscana.it (G.B.); 4Department of Medical Biotechnologies, University of Siena, Viale Bracci 16, 53100 Siena, Italy; ilariavicenti@gmail.com (I.V.); dragonifilippo@gmail.com (F.D.); 5VisMederi S.r.l., Strada del Petriccio e Belriguardo 35, 53100 Siena, Italy; 6Department of Medical Sciences, Surgery and Neurosciences, University of Siena, Viale Bracci 16, 53100 Siena, Italy; pierleopoldo.capecchi@unisi.it

**Keywords:** influenza A and B viruses, severe acute respiratory infections, epidemiological and virological surveillance

## Abstract

In Italy, the influenza season lasts from October until April of the following year. Influenza A and B viruses are the two viral types that cocirculate during seasonal epidemics and are the main causes of respiratory infections. We analyzed influenza A and B viruses in samples from hospitalized patients at Le Scotte University Hospital in Siena (Central Italy). From 2015 to 2020, 182 patients with Severe Acute Respiratory Infections were enrolled. Oropharyngeal swabs were collected from patients and tested by means of reverse transcriptase-polymerase chain reaction to identify influenza A(H3N2), A(H1N1)pdm09 and B. Epidemiological and virological surveillance remain an essential tool for monitoring circulating viruses and possible mismatches with seasonal vaccine strains, and provide information that can be used to improve the composition of influenza vaccines.

## 1. Introduction

Influenza has always been one of the most common respiratory diseases in the world. There are four types of influenza viruses: A, B, C, and D. Type A viruses are classified into subtypes based on their two surface proteins, hemagglutinin (HA) and neuraminidase (NA). Type A influenza viruses normally cause seasonal epidemics; they can also cause pandemics, the last one being caused by the H1N1 A/California/07/2009 A(H1N1)pdm09 virus in 2009. Type B viruses, which cause epidemic disease in humans, are not classified into subtypes; rather, they are usually broken down into lineages: B/Yamagata and B/Victoria. Type C causes respiratory infections with very mild symptoms [1]. Although influenza D virus primarily affects cattle and is not known to infect or cause disease in humans, recent research has shown the presence of antibodies in human sera [2].

Virological surveillance of influenza is important in order to determine the timing and spread of influenza viruses [3] and track changes in circulating influenza viruses, so as to inform seasonal influenza vaccine composition [4].

Influenza surveillance in Europe and in Italy is implemented by primary-care sentinel sites, which collect specimens from patients with influenza-like illness (ILI) and/or acute respiratory infection (ARI). In Italy, the surveillance system was founded in 1996 [5]. However, only during the 2009 influenza pandemic did the need for more global data on severe influenza disease starkly emerge, prompting the World Health Organization (WHO) to recommend conducting surveillance for hospitalized severe acute respiratory infection (SARI). This surveillance aims to collect information on severe and complicated forms of influenza and related deaths, in order to better understand the epidemiology of SARI in Italy [6]; indeed, before 2011, a global-surveillance case-definition of SARI did not exist [7].

In Italy, InfluNet is a national system for the virological and epidemiological surveillance of influenza. Coordinated by the National Center for Influenza of the Italian National Public Health Institute (Istituto Superiore della Sanità) in Rome, it enlists the collaboration of general practitioners, pediatricians and regional reference laboratories of the InfluNet network [8], in order to monitor the circulation of influenza viruses every winter, from the 42nd (mid-October) to the 17th week of the following year (late April) [9]. The Italian Ministry of Health recommends that the monitoring of SARI be widely implemented in the intensive care units (ICU) of local hospitals, and has requested their compliance [10]. In Italy, the InFluNet network is integrated by FluNews, which collects the results of several influenza surveillance systems, namely InfluWeb, which records the spontaneous reports of citizens on a website, providing a picture of the geographic distribution of influenza; InfluNet-Epi, an epidemiological surveillance system based on reports from general practitioners and pediatricians; and InfluNet-Vir, a virological system based on samples sent by general practitioners, pediatricians and hospitals to regional reference laboratories for the surveillance of SARI [11].

All these kinds of reporting are very important, as they provide as real as possible a picture of influenza from both the epidemiological and virological standpoints. Moreover, the reporting of SARI also extends to concomitant chronic diseases, as these can lead to an inauspicious outcome. Indeed, in a person with chronic diseases, the complications of influenza, such as viral and bacterial pneumonia, should not be underestimated [12]. The consequences of influenza infection can be severe, both for individuals and for the healthcare system. The severity of the infection depends on the type/subtype of the virus and the characteristics of the host (e.g., age), in particular, complications of influenza, such as pneumonia, are more common among specific risk groups, including the elderly, infants under one year of age, and subjects with immune deficiencies. SARI caused by the influenza virus can result in hospitalization [13].

In this study, we described the circulation of influenza viruses in the hospital setting in adults and elderly patients with SARI from the 2015/2016 to the 2019/2020 season, analyzed the virological characteristics of the strains detected and observed possible differences among the strains isolated.

## 2. Materials and Methods

### 2.1. Study Design

Oropharyngeal swabs were collected at the Unit of Emergency Medicine and Internal Medicine II of Le Scotte University Hospital in Siena, Italy, from the 2015 to the 2020 influenza seasons; specifically, sample collection was conducted in the context of the projects I-MOVE+ from 2015 to 2018 and DRIVE in the 2018–2020 influenza seasons. Both studies were approved by the Ethics Committee of Area Vasta Sud Est of Tuscany: approval Report of 16 November 2015, 21 November 2016, 18 December 2017, 22 January 2019, n.16344 on 16 December 2019 and n.18406 on 19 October 2020. Written informed consent was obtained from all patients included.

Oropharyngeal swabs were collected from hospitalized patients who fulfilled at least two enrolment criteria, i.e., at least one systemic symptom (fever or feverishness, headache, myalgia, generalized malaise) or deterioration of general conditions and at least one respiratory sign or symptom (cough, sore throat, breathing difficulties), present at the time of admission or within 48 h after admission to the hospital [14]. The starting date of symptoms (or aggravation of the basic conditions, if chronic) must not exceed 7 days prior to hospital admission. During interviews, patients were asked about their vaccination status. Each patient’s general practitioner (GP) was asked to confirm the vaccination status and the type of vaccine (trivalent or quadrivalent: TIV or QIV). Patients were included if they had been vaccinated more than 14 days before the onset of SARI symptoms. During patient interviews, the following underlying conditions were recorded: lung disease, heart disease, diabetes, renal disease, diseases of the hematopoietic organs and hemoglobinopathies, cancer, liver disease, obesity, anemia and enlarged spleen, leukemia, lymphomas, nutritional deficiency, dementia or stroke, rheumatologic disease, congenital and acquired diseases involving deficient antibody production, immunosuppression due to drugs or human immunodeficiency virus, chronic inflammatory diseases and intestinal malabsorption syndromes, and diseases associated with an increased risk of aspiration of respiratory secretions.

From 2015 to 2020, a total of 182 swabs were collected: season 2015/2016 (*n* = 41), season 2016/2017 (*n* = 19), season 2017/2018 (*n* = 17), season 2018/2019 (*n* = 38), season 2019/2020 (*n* = 67).

The median age of subjects was 80 years (range 47–99 years) and 51.6% were male. Out of 182 subjects, 66 (36.3%) had received seasonal influenza vaccination. The type of vaccine received was known only for 48 subjects, of which 45 (68.2%) and (4.5%) received TIV and QIV, respectively. For the remaining 18 subjects (27.3%) the type of vaccine was not known. The prevalence of underlying conditions is shown in Table 1.

### 2.2. Laboratory Analysis 

Total RNA was extracted from the oropharyngeal swabs by means of the QIAamp Viral RNA Mini kit (Qiagen, Hilden, Germany). 

One-step real time RT-PCR was performed in a final volume of 25 µL with 0.8 µM forward and reverse primers, 0.2 µM probe and 5 µL of extracted RNA following the manufacturer instruction included in the One-Step RT-PCR Kit (SuperScript III Platinum One-Step qRT-PCR Kit, Thermo Fisher Scientific, Waltham, MA, USA). Cycling conditions were 50 °C for 30 min, 95 °C for 2 min and 45 cycles of 15 s at 95 °C and 30 s at 55 °C. Fluorescence was measured during the 55 °C annealing/extension step (according to the Centers for Disease Control and Prevention—Influenza Division (CDC), Atlanta, GA, USA).

### 2.3. Sequencing Methods

Bidirectional DNA sequencing reactions were performed by means of the BrilliantDye Terminator Kit v1.1 (NimaGen, Nijmegen, The Netherlands) with six different primers for A(H1N1) and four different primers for A(H3N2), spanning the HA gene of interest. Briefly, 3 μL of PCR products, diluted to a final concentration of 1–3 ng/μL, were mixed with 3.2 pmol/μL of each sequencing primer, 0.5 μL of BrilliantDye Terminator Ready Reaction Sequencing and 2 μL of 5× Sequencing Buffer in a final volume of 10 μL. The reactions were denatured at 96 °C for 1 min, followed by 25 cycles at 50 °C for 5 s, 60 °C for 4 min and 96 °C for 10 s. Sequencing reactions were treated with the X-Terminator Purification kit (Applied Biosystems, Foster City, CA, USA) in a 96-well plate, as suggested by the manufacturer, then resolved by capillary electrophoresis with the 3130 XL Genetic Analyzer (Applied Biosystems, Foster City, CA, USA). Chromatograms were assembled and edited by means of the DNAStar 7.1.0 SeqMan module (DNASTAR, Madison, WI, USA).

### 2.4. Influenza Hemagglutinin Multiple-Sequence Alignment

Multiple-sequence alignment was performed by means of the Basic Local Alignment Search Tool (BLAST) server. HA sequences of swabs positive for A(H3N2) in the 2016/2017 season and for A(H1N1)pdm09 in the 2017/2018 season were compared with HA sequences of reference vaccine strains for the respective seasons of isolation (A/Hong Kong/4801/2014, 2016/2017 season and A/Michigan/45/2015, 2017/2018 season). 

### 2.5. Statistical Analysis

The median ages of both the study population and positive subjects were calculated. Prevalence rates were calculated, together with their corresponding 95% confidence intervals (CI) and compared by means of Yates’ corrected chi-square test. Statistical significance was set at *p* < 0.05, two-tailed. All statistical analyses were performed by means of GraphPad Prism 6 software.

## 3. Results

Out of a total of 182 subjects recruited over five influenza seasons (2015/2016, 2016/2017, 2017/2018, 2018/2019, 2019/2020), 30 (16.5%, 95% CI 11.7–22.6) were laboratory-confirmed influenza-positive cases. One case of coinfection with influenza A(H1N1)pdm09 and A(H3N2) was identified in the 2017/2018 season. All B viruses belonged to the Yamagata lineage. Table 2 shows positive swabs by influenza (sub)type and season.

As shown in Figure 1, A(H3N2) was more frequently found than A(H1N1)pdm09 and B viruses (*p* = 0.01 and *p* < 0.0001, respectively). 

The median age of positive subjects was 78 years (range 56–90 years), and 50.0% were male; 66.7% (95% CI 47.2–82.7) of positive subjects had not received any influenza vaccine for the season; of the 33.3% (95% CI 19.1–51.3) who had been vaccinated, 9 (30.0%, 95% CI 14.7–49.4) had received TIV, and 1 (3.3%, 95% CI 0.1–17.2) QIV. No significant differences were found in terms of vaccination status among positive subjects or in comparison with negative ones.

Positive subjects had a mean of 2.2 (±1.4) underlying conditions; heart and lung diseases were the most frequently reported, present in 53.3% (95% CI 36.1–69.8) and 46.7% (95% CI 30.2–63.9) of cases, respectively, followed by renal disease and cancer (26.7%, 95% CI 14.0–44.7), and diabetes (20.0%, 95% CI 9.1–37.7). With regard to underlying conditions, no significant differences were found among positive subjects.

During the 2017/2018 season, a case of coinfection with influenza viruses A(H1N1)pdm09 and A(H3N2) was identified in a 68-year-old female, who had been vaccinated with QIV. She had all the systemic symptoms (fever, malaise, headache, and myalgia) and local symptoms (cough, sore throat, and shortness of breath) of SARI, and suffered from several chronic diseases (cancer, anemia, leukemia and a disease involving deficient antibody production). 

Isolates from swabs positive for A(H1N1)pdm09 and A(H3N2) in the 2016/2017 and 2017/2018 seasons were sequenced and compared with reference vaccine strains for the season in question.

One A(H3N2) isolate from the 2016/2017 season belonged to the genetic group 3C.2a; the other 4 isolates from the same season converged within the more recent subclade 3C.2a1, which is defined by the further amino acid substitutions N171K, I406V and G484E in the HA gene. A(H3N2) isolate from the 2017/2018 season also showed the amino acid substitution I406V. A(H1N1) isolate from the 2017/2018 season belonged to the genetic group 6B.1, which is defined by additional amino acid substitutions S74R, S164T, I295V, and T120A. 

## 4. Discussion

In this study, we found that 16.5% of 182 subjects hospitalized in Siena, Tuscany, from 2015 to 2020 with SARI symptoms were positive for influenza virus infection. Most of the infections were sustained by type A viruses, especially A(H3N2) viruses, which accounted for two-thirds of the infections in our study. These values are in line with the trend reported by virological surveillance in Italy and in Europe [15,16,17,18,19,20]. In Italy, average vaccination coverage from 2015 to 2020 was 52.46% in subjects aged over 65 years and 15.38% in the general population [21].

During the 2015/2016 influenza season, virus type A(H1N1)pdm09 initially predominated in both Europe and Italy [22], while virus type B (mainly Victoria lineage) was slightly more prevalent than type A(H1N1)pdm09 at the end of the season [22]. Virological surveillance reported that most A(H1N1) viruses detected clustered in a new genetic subclade 6B.1, which was antigenically like the vaccine component A/California/7/2009. In addition, the surveillance data showed a predominance of B virus (Victoria lineage) distinct from the Yamagata vaccine component [23].

In the 2016/2017 influenza season, A(H3N2) viruses largely predominated in Italy, and the results of virological surveillance in Europe confirm this trend [16]. Of the five A(H3N2) viruses that we isolated, four converged in the subclade 3C.2a1, like most of the A(H3N2) viruses isolated in Italy and in the world in the same season. These isolates were collected from subjects who had been vaccinated in that influenza season; indeed, the infection was sustained by a strain that did not match the vaccine strain A(H3N2) A/Hong Kong/4801/2014 [15,16,24]. One A(H3N2) isolate from the 2016/2017 season belonged to the genetic group 3C.2a, which also includes the vaccine strain A/Hong Kong/4801/2014 in that season. This isolate was collected from a subject who had not undergone influenza vaccination.

During the 2017/2018 season, influenza B (Yamagata lineage) viruses were dominant and A(H1N1) and A(H3N2) cocirculated, but the pattern and magnitude of their circulation varied across countries; in Italy, A(H1N1) was particularly dominant. Towards the end of surveillance, we found two patients positive for influenza B virus belonging to the Yamagata lineage that was not included in the TIV. These two subjects had not been vaccinated. This finding is in line with other studies [17,25] and highlights the fact that the circulating B virus was different from the one included in the TIV, leading to a mismatch. Nevertheless, the influenza vaccine effectiveness (IVE) against B viruses was estimated to be moderate; this could be explained by the fact that most of the population had received the adjuvanted TIV, which was able to confer a cross-lineage protection [17,25].

During surveillance of the 2017/2018 season, we identified a case of coinfection with influenza viruses A(H1N1)pdm09 and A(H3N2). The A(H1N1) isolate belonged to the genetic group 6B.1, to which the vaccine strain A/Michigan/45/2015 also belongs, and displayed the additional amino acid substitution that characterized most of the strains isolated in Italy during the same season [25]. In the 2017/2018 season, the A/Michigan/45/2015 (H1N1)pdm09-like virus vaccine component replaced the A/California/7/2009 (H1N1)pdm09-like virus vaccine component, which had been recommended for seven consecutive years, from 2010/2011 to 2016/2017. The A(H3N2) isolate belonged to the subclade 3C.2a1, like most of the A(H3N2) viruses we had isolated in the previous season. This isolate was also collected from a subject who had received the vaccine for the influenza season in question, suggesting a mismatch with the circulating strain.

In Europe, most of the circulating viruses that were analyzed were antigenically similar to the vaccine component A(H3N2) clade 3C.2. Nevertheless, one third of the viruses isolated had undergone a genetic change, resulting in a 3C.2.a1 subclade, like the five isolates in our study. In the same season, most of the A(H3N2) viruses circulating in Europe belonged to the clade 3C.2 and were antigenically similar to the vaccine strain (A/Hong Kong/4801/2014). However, one third of the influenza A(H3N2) viruses sequenced belonged to the subclade 3C.2a1 [17], as did most of the viruses isolated in our study. For this reason, in September 2017, the WHO was prompted to replace the vaccine component with A/Singapore/INFIMH-16-0019/2016 (subclade 3C.2a1) in the 2017/2018 season in the southern hemisphere, and subsequently in the northern hemisphere for the 2018/2019 season [26].

In the 2018/2019 season, A(H3N2) and A(H1N1)pdm09 viruses cocirculated. With regard to A(H1N1)pdm09, there was a good match between the circulating and vaccine strains. However, the WHO decided to change the subtype of A/H3N2 in the vaccine composition for the next influenza season, as the wild virus A/H3N2 was antigenically different from the strain included in the 2018/2019 vaccine [26]. In 2019/2020, A(H1N1) virus and, particularly, A(H3N2) virus were isolated. During this season, too, the mismatch between the circulating A/H3N2 virus and the vaccine strain prompted the WHO to substitute the vaccine component for the 2020/2021 season [27].

Our study has some limitations. First, as the sample size was limited by the overall availability of swabs collected, it may not have been fully representative of the population. Second, isolates from only two influenza seasons were sequenced, and as sequencing of A(H3N2) isolates from the 2017/2018 season was partial, the sequence analysis may have been incomplete.

## 5. Conclusions

Overall, our data support the importance of seasonal vaccination in subjects with chronic diseases and highlight the key role of epidemiological and virological surveillance as an essential tool for monitoring circulating viruses and possible mismatches with seasonal vaccine strains, and providing information that can be used to improve the composition of influenza vaccines.

## Figures and Tables

**Figure 1 ijerph-18-03875-f001:**
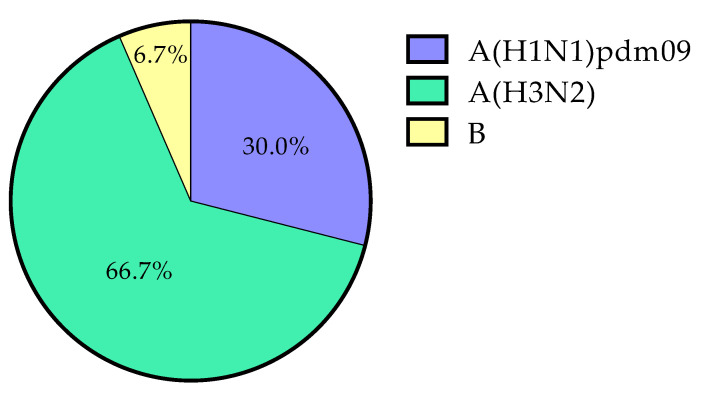
Prevalence by (sub)type: A(H1N1)pdm09, *n* = 9 (30.0%, 95% CI 16.5–48.0); A(H3N2), *n* = 20 (66.7%, 95% CI 48.7–80.9); B, *n* = 2 (6.7%, 95% CI 0.8–22.4).

**Table 1 ijerph-18-03875-t001:** Underlying conditions in the study population: number of subjects (N) and prevalence (%).

Underlying Conditions	N	%
Lung disease	118	64.8
Heart disease	112	61.5
Diabetes	65	35.7
Renal disease	53	29.1
Diseases of the hematopoietic organs and hemoglobinopathies	18	9.9
Cancer	35	19.2
Liver disease	8	4.4
Obesity	17	9.3
Anemia and enlarged spleen	24	13.2
Leukemia, lymphoma	6	3.3
Nutritional deficiency	5	2.7
Dementia or stroke	25	13.7
Rheumatologic disease	17	9.3
Congenital and acquired diseases involving deficient antibody production	8	4.4
Immunosuppression due to drugs or HIV	8	4.4
Chronic inflammatory diseases and intestinal malabsorption syndromes	10	5.5
Diseases associated with an increased risk of aspiration of respiratory secretions	12	6.6

**Table 2 ijerph-18-03875-t002:** Positive swabs by influenza (sub)type and season, total and prevalence (%) (95% CI); * case of coinfection.

Season	A(H1N1)pdm09	A(H3N2)	B	Total	% (95% CI)
2015/2016	1	0	0	1	2.4% (0.0–13.7)
2016/2017	0	5	0	5	26.3% (11.4–49.1)
2017/2018	1 *	1 *	2	3	17.6% (9.0–47.8)
2018/2019	2	6	0	8	21.1% (10.8–36.6)
2019/2020	5	8	0	13	19.4% (11.6–30.6)

## Data Availability

Data sharing not applicable No new data were created or analyzed in this study.

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
