# Peer review of "Surveillance for Severe Acute Respiratory Infections among Hospitalized Subjects from 2015/2016 to 2019/2020 Seasons in Tuscany, Italy"

_ijerph, 2021, doi:10.3390/ijerph18083875_

Round 1

Reviewer 1 Report

The authors analyze the influenza A and B viruses in samples from  hospitalized patients at Le Scotte University Hospital in Siena (Central Italy). Although authors paid special attention on explaining and analyzing virological characteristics of the strains detected in patients, the study has serious limitations, such as the low number of evaluated samples (30). In addition, isolates from only two influenza seasons were sequenced, making the analysis incomplete. Therefore a new version of the manuscript with substantial changes is advisable before the article can be considered for publication: The authors have to consider the analysis of more samples (consider another season, for instance 2014-2015).

Other suggestions are described below.

- Introduction:

- lines 45-47: The authors should add at least one reference of an example where tracking of changes in circulating influenza viruses was possible through a virological surveillance.

-Lines 79-83: Rewrite these sentences. The indication of the factors related to a severe and complicated influenza infection is repetitive.

Results:

-Table 2: The heading in column 5 should be "N" instead of "Total", as indicated in the table description.

Author Response

REVIEWER 1:

Comments and Suggestions for Authors

The authors analyze the influenza A and B viruses in samples from hospitalized patients at Le Scotte University Hospital in Siena (Central Italy). Although authors paid special attention on explaining and analyzing virological characteristics of the strains detected in patients, the study has serious limitations, such as the low number of evaluated samples (30). In addition, isolates from only two influenza seasons were sequenced, making the analysis incomplete. Therefore a new version of the manuscript with substantial changes is advisable before the article can be considered for publication: The authors have to consider the analysis of more samples (consider another season, for instance 2014-2015).

Response:

Thanks for your suggestions. As regards the analysis of several samples, all available samples were analyzed.

Other suggestions are described below.

- Introduction:

- lines 45-47: The authors should add at least one reference of an example where tracking of changes in circulating influenza viruses was possible through a virological surveillance.

Response:

The reference has been added.

-Lines 79-83: Rewrite these sentences. The indication of the factors related to a severe and complicated influenza infection is repetitive.

Response:

These sentences have been rephrased (lines 79-84).

Results:

-Table 2: The heading in column 5 should be "N" instead of "Total", as indicated in the table.

Response:

The heading in column has been edited.

Reviewer 2 Report

The authors should indicate out of 182 patients how many were vaccinated for Influenza and then compare Flu negative patients against Flu positive patients to make a conclusion that seasonal vaccination is recommended based on their study.

Line 125: Should state that "Presumptive viral RNA or nucleic acid was extracted from the samples using the kit not Influenza A or B viral RNA". Please rewrite the sentence.

Author Response

REVIEWER 2:

Comments and Suggestions for Authors

The authors should indicate out of 182 patients how many were vaccinated for Influenza and then compare Flu negative patients against Flu positive patients to make a conclusion that seasonal vaccination is recommended based on their study.

Response:

Thanks for the suggestion. The number of subjects who received influenza vaccination has been added (lines 121-124) and comparison between influenza negative and positive subjects has been performed (lines 181-183).

Line 125: Should state that "Presumptive viral RNA or nucleic acid was extracted from the samples using the kit not Influenza A or B viral RNA". Please rewrite the sentence.

Response:

The sentence has been rephrased (lines 129).

Reviewer 3 Report

The manuscript by Manini et al. presents surveillance data for influenza infections in seasons 2015/2016 to 2019/2020. The authors present important data comparing circulating strains of influenza virus with the administered vaccine strains. The study is limited with regard to sample size, but this shortcoming is clearly mentioned and as such should not prevent publication of this study.

Another weak point of the manuscript is lack of the sequencing data for all of the positive samples. It might be a good idea to ask the authors to go back and sequence all of the samples, however I am not sure if this will affect the conclusions significantly.  

Author Response

Comments and Suggestions for Authors

The manuscript by Manini et al. presents surveillance data for influenza infections in seasons 2015/2016 to 2019/2020. The authors present important data comparing circulating strains of influenza virus with the administered vaccine strains. The study is limited with regard to sample size, but this shortcoming is clearly mentioned and as such should not prevent publication of this study.

Another weak point of the manuscript is lack of the sequencing data for all of the positive samples. It might be a good idea to ask the authors to go back and sequence all of the samples, however I am not sure if this will affect the conclusions significantly.

Response:

Thanks for your suggestion, however, currently we have no chance to sequence all of the samples.

Round 2

Reviewer 1 Report

The authors addressed the minor suggestions. However the major issue about considering the analysis of more samples was not resolved. In my opinion the manuscript without this correction does not fulfill the journal standards. For this reason I recommend to reject it.

Author Response

REVIEWER 1:

Comments and Suggestions for Authors

The authors addressed the minor suggestions. However the major issue about considering the analysis of more samples was not resolved. In my opinion the manuscript without this correction does not fulfill the journal standards. For this reason I recommend to reject it.

Response:

Thanks for your suggestion. Unfortunately the sample size cannot be increased since these samples were collected for the studies described in the manuscript and we have already analyzed all the available samples (182), in particular 41 samples were collected in 2015/2016 season, 19 samples in 2016/2017 season, 17 samples in 2017/2018 season, 38 samples in 2018/2019 season and 67 samples in 2019/2020.We are aware that our study is limited by the sample size and we clearly mentioned it as a limitation of the study (lines 265-266). We hope this will be acceptable.